Structural analysis of M1AP variants associated with severely impaired spermatogenesis causing male infertility

http://orcid.org/0000-0003-4288-6340 Gerlevik Umut 1 2
Ergoren Mahmut Cerkez 3 4
Sezerman Osman Uğur 1 5 ugur.sezerman@acibadem.edu.tr
Temel Sehime Gulsun 6 7 8 sehime@uludag.edu.tr
1 Department of Biostatistics and Bioinformatics, Institute of Health Sciences, Acibadem Mehmet Ali Aydinlar University , Istanbul , Turkey
2 Department of Biochemistry, University of Oxford , Oxford , United Kingdom
3 Department of Medical Genetics, Faculty of Medicine, Near East University , Nicosia , Cyprus
4 DESAM Institute, Near East University , Nicosia , Cyprus
5 Department of Biostatistics and Medical Informatics, School of Medicine, Acibadem Mehmet Ali Aydinlar University , Istanbul , Turkey
6 Department of Medical Genetics, Faculty of Medicine, Bursa Uludag University , Bursa , Turkey
7 Department of Histology & Embryology, Faculty of Medicine, Bursa Uludag University , Bursa , Turkey
8 Department of Translational Medicine, Health Sciences Institute, Bursa Uludag University , Bursa , Turkey
Uversky Vladimir
Electronic publication date: 2022 Mar 21
Publication date: 2022
Volume: 10
Electronic Location ID: e12947
Received 2021 Mar 12; Accepted 2022 Jan 25
Copyright: © 2022 Gerlevik et al.
Copyright year: 2022
Copyright holder: Gerlevik et al.
License: This is an open access article distributed under the terms of the Creative Commons Attribution License, which permits unrestricted use, distribution, reproduction and adaptation in any medium and for any purpose provided that it is properly attributed. For attribution, the original author(s), title, publication source (PeerJ) and either DOI or URL of the article must be cited.
License URL: https://creativecommons.org/licenses/by/4.0/

Keywords: Molecular modelling, Molecular dynamics simulations, Male infertility, Meisos 1-associated protein (M1AP), Non-obstructive azoospermia (NOA), Cryptozoospermia, Variant effect on protein structure

Funding: The authors received no funding for this work.

==============================
Background

Impaired meiosis can result in absence of sperm in the seminal fluid. This condition, namely non-obstructive azoospermia (NOA), is one of the reasons of male infertility. Despite the low number of studies on meiosis 1-associated protein (M1AP) in the literature, M1AP is known to be crucial for spermatogenesis. Recently, seven variants (five missense, one frameshift, one splice-site) have been reported in the M1AP gene as associated with NOA, cryptozoospermia and oligozoospermia in two separate studies. However, all missense variants were evaluated as variant of uncertain significance by these studies. Therefore, we aimed to analyze their structural impacts on the M1AP protein that could lead to NOA.

Methods

We firstly performed an evolutionary conservation analysis for the variant positions. Afterwards, a comprehensive molecular modelling study was performed for the M1AP structure. By utilizing this model, protein dynamics were sampled for the wild-type and variants by performing molecular dynamics (MD) simulations.

Results

All variant positions are highly conserved, indicating that they are potentially important for function. In MD simulations, none of the variants led to a general misfolding or loss of stability in the protein structure, but they did cause severe modifications in the conformational dynamics of M1AP, particularly through changes in local interactions affecting flexibility, hinge and secondary structure.

Conclusions

Due to critical perturbations in protein dynamics, we propose that these variants may cause NOA by affecting important interactions regulating meiosis, particularly in wild-type M1AP deficiency since the variants are reported to be homozygous or bi-allelic in the infertile individuals. Our results provided reasonable insights about the M1AP structure and the effects of the variants to the structure and dynamics, which should be further investigated by experimental studies to validate.

Introduction

Male infertility is a widespread problem in human reproduction, affecting many people in the world (Agarwal et al., 2015; Olszewska et al., 2020). There are numerous conditions that cause male infertility, and two of them are azoospermia and cryptozoospermia (World Health Organization, 2010). Azoospermia is the condition of lacking spermatozoa in seminal fluid. Moreover, there are two main types of azoospermia: obstructive (normal spermatogenesis) and non-obstructive (damaged spermatogenesis) (Gudeloglu & Parekattil, 2013; Hwang et al., 2018). In other words, there is no problem in sperm formation in obstructive azoospermia but there are other problems after the formation. Contrarily, the main problem in non-obstructive azoospermia (NOA) is during spermatogenesis. On the other hand, cryptozoospermia is the condition where there are no spermatozoa in fresh preparations but very few spermatozoa in the pellet of the centrifuged semen sample (Zhu et al., 2016; Karabulut et al., 2018). Besides the clinical similarities between NOA and cryptozoospermia, they may also have common genetic basis based on problems of spermatogenesis (Arango et al., 2006, 2013; World Health Organization, 2010; Wyrwoll et al., 2020). Distorted spermatogenesis is mostly relied on a genetic origin, but the affected individuals are routinely screened for chromosomal abnormalities and Y chromosome azoospermia factor microdeletions (Lee et al., 2011). These approaches cover only up to 20% of the cases affected by azoospermia alone (Tüttelmann, Ruckert & Röpke, 2018). Recent studies indicated that human male germ cell arrest is caused by monogenic variations (Gershoni et al., 2017) such as testis-expressed gene 11 (TEX11, MIM 300311) gene on the X chromosome (Yatsenko et al., 2015). Furthermore, bi-allelic genetic variants on meiosis 1-associated protein (M1AP, MIM 619098) gene were recently associated with both of NOA and cryptozoospermia in different patients (Wyrwoll et al., 2020). Thus, in these cases, NOA and cryptozoospermia could be closely associated in terms of the underlying mechanism (Wyrwoll et al., 2020).

M1AP is a gene expressed in germ cells in both males and females, and it encodes a 530 amino acid long protein (M1AP). Arango et al. (2006) found that this gene is expressed at the beginning of meiosis in female germ cells and at the final stages of spermatogenesis in male germ cells. Therefore, they suggest that the evolutionarily well-conserved M1AP protein may have a role in gametogenesis. Furthermore, Arango et al. (2013) showed that M1AP-deficient male mice have severe oligozoospermia and infertility with many testicular defects. Therefore, they suggest that mutations in M1AP might be reason for nonobstructive oligozoospermia in men. In addition, they localized M1AP as a cytosolic protein. Unfortunately, no further knowledge is available in the literature due to the lack of focus on this protein, which has the potential to be highly critical for meiosis and reproduction.

To the best of our knowledge, 1 frameshift, 1 splice site and 5 missense variants found in M1AP have so far been associated with male infertility conditions (i.e., NOA, cryptozoospermia and oligozoospermia) in previous studies (Tu et al., 2020; Wyrwoll et al., 2020). As shown in Table 1, all these variants except the homozygous splice site variant were found in different bi-allelic combinations in the individuals rather than single variants (Wyrwoll et al., 2020). In addition, a fertile person was found despite having the heterozygous p.Pro389Leu variant (Wyrwoll et al., 2020). On the other hand, since the variants are on different alleles (compound heterozygous), any two of them are not expected to be in a protein product at the same time although they could be found in the same cell as separate proteins. Even though the algorithms predict the missense variants as mostly damaging (Table 1), Wyrwoll et al. concluded these variants individually as variants of uncertain significance based on a classification according to the ACMG-AMP Guidelines (Richards et al., 2015).

Table 1 Bi-allelic M1AP variants reported in infertile males and predictions for their functional effects by Wyrwoll et al. (2020).

Individual	Bi-allelic M1AP variants	Variant impact predictions
(PolyPhen-2/SIFT/MutationTaster)	Diagnosis	
	Allele 1	Allele 2	Allele 1	Allele 2		
P86	p.Ser50Pro	p.Leu430Pro	Tolerated/Polymorphism/Damaging	Damaging/Damaging/Damaging	Azoospermia	
M1943	p.Arg266Gln	p.Trp226LeufsTer4	Damaging/Damaging/Damaging	–	Cryptozoospermia/
azoospermia†	
Y126	p.Gly317Arg	p.Trp226LeufsTer4	Damaging/Damaging/Damaging	–	Azoospermia	
T1024	p.Pro389Leu	p.Pro389Leu	Damaging/Damaging/Damaging	Damaging/Damaging/Damaging	Azoospermia	
Notes:

The missense variants in the table (i.e., p.Ser50Pro, p.Arg266Gln, p.Gly317Arg, p.Pro389Leu and p.Leu430Pro) were examined in this study.

All data in the table were reported in Wyrwoll et al. (2020). Although there were more individuals in the reference study, only bi-allelic variants with at least one missense change were included in this study since our focus is on the impacts of missense variants on the M1AP structure.

† Semen contained none or below 10 spermatozoa/sample on repeated analyses.

Although there is almost no information about M1AP structure and function in the literature, the effects of the missense variants might be revealed by molecular modeling and molecular dynamics (MD) simulations. These approaches allow to assess the impacts on protein structure and dynamics. Owing to its robustness, computational modeling is well-suited for studying missense variants associated with genetic diseases (Zimmermann et al., 2017). In this study, we investigated the structural mechanisms underlying NOA and cryptozoospermia by analyzing the impacts of five missense variants on the structure and dynamics of M1AP via MD simulations.

Methods

A comprehensive whole-exome sequencing and variant prioritization study (Wyrwoll et al., 2020) has previously shown that some biallelic missense variants in the meiosis 1-associated protein (M1AP) gene were associated with male infertility via non-obstructive azoospermia (NOA) or cryptozoospermia. Each missense variant shown in Table 1 was analyzed individually in this study for their impacts on M1AP structure and dynamics.

Evolutionary conservations

It is known that structurally and functionally important amino acids are mostly conserved during evolution (Ashkenazy et al., 2016). To reveal the importance of variant regions, we aligned the UniProt (The UniProt Consortium, 2019) protein sequences of M1AP orthologs from human (Homo sapiens, UniProt ID: Q8TC57), chimpanzee (Pan troglodytes, UniProt ID: H2R3U8), cat (Felis catus, UniProt ID: A0A337RXR8), wild boar (Sus scrofa, UniProt ID: A0A5G2QVF2), bovine (Bos taurus, UniProt ID: E1BF42) and house mouse (Mus musculus, UniProt ID: Q9Z0E1) by using T-Coffee with its default parameters (Notredame, Higgins & Heringa, 2000; Di Tommaso et al., 2011). The alignment was visualized via ESPript (v3.0) (Robert & Gouet, 2014).

Molecular modeling and validation

Sequence of the canonical isoform of M1AP protein (UniProt ID: Q8TC57-1) was used to build models. GalaxyWeb (Ko et al., 2012), I-TASSER (Roy, Kucukural & Zhang, 2010), Phyre2 (Kelley et al., 2015), PRIMO (Hatherley et al., 2016), RaptorX (Peng & Xu, 2011) and Robetta (Kim, Chivian & Baker, 2004) were used to model different structures with their default parameters. Regular comparative algorithm (we called Robetta model) and domain prediction algorithm (we called Robetta-domain model), which are part of Robetta webserver (https://robetta.bakerlab.org), were applied separately (Kim, Chivian & Baker, 2004). Robetta algorithms also applied an ab initio prediction for part of the structure and compared their reliability with its template-based predictions.

A list of the templates automatically selected by the modeling algorithms, along with their experimental methodologies, resolution values, and Ramachandran outliers are available in Table S1. Since all structure predictors do not list the identity, similarity, and coverage details, we aligned the templates with M1AP via NCBI’s Protein BLAST (https://blast.ncbi.nlm.nih.gov/) by applying default parameters except the removed E-value threshold (Table S2).

Quality of the structures were assessed by Verify3D (Eisenberg, Lüthy & Bowie, 1997), ERRAT (Colovos & Yeates, 1993), Prove (Pontius, Richelle & Wodak, 1996), PROCHECK (Laskowski et al., 1993) and WHATCHECK (Hooft et al., 1996) algorithms on SAVES v6.0 webserver (https://saves.mbi.ucla.edu/). Using these assessments, we firstly selected the best model from each algorithm and then compared them with each other. Since the qualities of the two best models were so close, they were further evaluated for folding stability via molecular dynamics (MD) simulations. Of note, before the quality assessments, the models subjected to MD simulations (see “further models” and “variant models” in Table 2) were processed with “RepairPDB” command of FoldX (v5.0) (Delgado et al., 2019; van Durme et al., 2011), which converts the saved coordinates to the PDB format of SAVES and applies a basic minimization procedure.

Table 2 Quality assessments of M1AP models from five different algorithms.

Initial and comparison models were not assessed after being processed with molecular dynamics (MD) simulations whereas further and variant models were assessed after being subjected to MD simulations for a time period specified at the end of the model name (e.g., 450ns).

Model	VERIFY3D2	ERRAT3	PROVE4	PROCHECK5	WHATCHECK6	
Aa1% with score ≥0.2	AOQF1%	BOPA1%	Aa% in MFR1	Aa% in AAR1	Aa% in GAR1	Aa% in DR1	G-Factor	# of SCP1 with pass	# of SCP with warning	# of SCP with error	Overall report	
Initial Models	
GalaxyWeb	69.25	74.07	8.1	89.1	8.4	1.7	0.8	–0.05	26	15	4	Pass	
I-TASSER	47.36	90.47	7.5	60.8	31.7	4.7	2.8	–0.77	19	18	10	Pass	
Phyre2	56.98	22.99	10.4	76.0	14.6	5.1	4.3	–0.60	27	12	7	Pass	
PRIMO	39.66	1.35	33.4	58.4	25.1	9.2	7.3	–1.41	23	15	8	Pass	
RaptorX	41.32	43.72	9.1	78.6	17.8	1.9	1.7	–0.42	23	16	7	Pass	
Robetta	76.23	95.22	4.8	83.9	15.6	0.3	0.2	0.11	27	14	6	Pass	
Robetta-domain	89.06	88.24	4.4	85.2	13.9	0.5	0.4	0.20	27	15	5	Pass	
Comparison Model	
AlphaFold2	89.55	63.21	4.4	80.9	14.3	3.5	1.3	–0.14	22	17	6	Pass	
Further Models	
GalaxyWeb_100ns	88.30	78.35	7.1	84.6	13.7	1.1	0.6	–1.11	17	21	8	Pass	
Robetta-domain_100ns	83.89	87.53	6.6	87.1	10.7	1.3	0.9	–1.11	17	21	8	Pass	
GalaxyWeb_450ns	80.75	85.38	5.9	89.1	9.4	1.3	0.2	–1.05	17	22	7	Pass	
Variant Models	
GalaxyWeb_Ser50Pro_450ns	79.62	87.45	8.5	88.0	9.2	2.4	0.4	–1.10	16	22	8	Pass	
GalaxyWeb_Arg266Gln_450ns	86.60	77.26	6.0	84.6	12.6	1.9	0.9	–1.14	16	20	10	Pass	
GalaxyWeb_Gly317Arg_450ns	84.91	74.90	7.1	85.5	12.4	1.1	1.0	–1.09	18	20	8	Pass	
GalaxyWeb_Pro389Leu_450ns	77.36	70.50	8.2	84.2	13.0	1.7	1.1	–1.11	16	21	9	Pass	
GalaxyWeb_Leu430Pro_450ns	79.06	78.46	6.3	85.6	12.0	1.7	0.7	–1.09	18	20	8	Pass	
Notes:

1 Aa, amino acid; AOQF, average overall quality factor; BOPA, buried outlier protein atoms; MFR, the most favored regions; AAR, additional allowed regions; GAR, generously allowed regions; DR, disallowed regions; SCP, stereochemical properties.

2 In VERIFY3D, at least 80% of aa in a protein should have a score ≥0.2 in the 3D–1D profile for a good quality.

3 In ERRAT, AOQF ≥95% indicates a good quality, AOQF around 91% indicates an average quality, and lower AOQF indicates a bad quality.

4 In PROVE, BOPA ≤1% indicates a good quality, BOPA between 1% and 5% indicates an average quality, and BOPA >5% indicates a bad quality.

5 In the Ramachandran plot analysis of PROCHECK, model has a good quality if overall G-factor is >–0.5 and ≥90% of aa in protein are in the most favored regions.

6 In WHATCHECK, SCP with pass refers to a good quality, SCP with warning indicates an average quality, and SCP with error refers to a bad quality. Of note, total number of SCP might vary from structure to structure.

During the revision process of this article, a revolution in protein structure prediction was carried out in the Critical Assessment of protein Structure Prediction (CASP) (Pereira et al., 2021) by Google DeepMind. They developed a deep learning-based algorithm called AlphaFold2 (AF2) which makes de novo structure prediction with high accuracy (Jumper et al., 2021; Thornton, Laskowski & Borkakoti, 2021; Varadi et al., 2021). Because of its success, we compared our model with the AF2 model as an additional validation. AlphaFold Protein Structure Database (https://alphafold.ebi.ac.uk/entry/Q8TC57) (Varadi et al., 2021) was used to obtain AF2’s M1AP model.

We used PyMOL’s iterative superposition algorithm (Schrödinger & DeLano, 2020) for the structural alignment of the AF2 model with our model (see Fig. S1) and to compare the models generated by the same homology modeling algorithm (see Fig. S2).

Visual Molecular Dynamics (VMD v1.9.3) (Humphrey, Dalke & Schulten, 1996) was used to model the structures with the variants via the VMD Mutator Plugin.

System preparation

PDB2PQR (Dolinsky et al., 2004) was used to calculate the protonation states of amino acids at pH 7. Moreover, Glu43, Glu436 and Asp118 in the GalaxyWeb models (i.e., wild-type and all mutants) were protonated whereas His185, His238 and His375 were protonated in the Robetta-domain model. Models were solvated in a TIP3P water (Jorgensen et al., 1983) box. The systems were neutralized and ionized by adding 0.15 M KCl. The dimensions of the system were approximately 112.6×108.8×125.2 Å3 in the xyz axes and the system contained ~146,000 atoms. VMD was used for the entire preparation procedure.

Simulation setup

NAMD 2.13-multicore-CUDA (Phillips et al., 2005) was used for MD simulations with the CHARMM36m all-atom force field (Huang et al., 2016). Simulations were performed under NPT ensemble with Langevin thermostat and Nosé-Hoover Langevin piston at 310 K and 1 atm (Feller et al., 1995; Davidchack, Handel & Tretyakov, 2009). The damping coefficient (gamma) of the Langevin thermostat was 1/ps. The oscillation period of the piston was around 100 fs whereas the oscillation decay time was 50 fs. A cutoff distance of 12 Å was used for the van der Waals interactions. The switching function starts at 10 Å and reaches zero at 14 Å. Long-range Coulomb interactions were calculated with particle-mash Ewald (Kolafa & Perram, 1992). A 2 fs integration time-step was used. The minimization and equilibration procedure was performed as follows: (1) 5,000 steps conjugate gradient algorithm and 1 ns equilibration of the solution by fixing the protein; (2) 5,000 steps of conjugate gradient minimization and 1 ns equilibration of the whole system by releasing all constraints except the ShakeH algorithm. This procedure provided sufficient relaxation for the system (Fig. S3). ShakeH of NAMD was applied during the entire simulation procedure to keep hydrogens constrained. Production simulations (Repeat 1) for each system, including wild-type and the variants, were performed for 500 ns. Production simulations were re-performed for 500 ns (Repeat 2) with different random number generator seeds to validate the reproducibility of the results and/or to obtain different possible conformations independent of the original simulations. In this way, the total production simulation time for each system was 1,000 ns. All simulation results are available at Zenodo (https://doi.org/10.5281/zenodo.5811977).

Structural and dynamical analyses

The “Generate” function of PDBsum (Laskowski, 2009) was used to predict the 2-D structure topology, clefts and tunnels of the model.

Alterations in a protein’s compactness, folding, solvent accessibility, stability, flexibility, hinges, local interactions (i.e., hydrogen bonds (H-bonds) and salt bridges within 10 Å of variant sites) and essential motions might influence functions, interactions and regulations of that protein. Therefore, we used in-house tcl scripts (available at https://github.com/ugerlevik/M1AP_analysis) and built-in VMD plugins to calculate backbone root-mean-square deviation (RMSD, related to folding), average Cα root-mean-square fluctuation (RMSF, indicates the flexibility), radius of gyration (Rg, inversely associated with compactness), solvent accessible surface area (SASA) and distances.

FoldX (v5.0) (Delgado et al., 2019) was used for stability analysis. ProDy (Bakan, Meireles & Bahar, 2011) was used for principal component analysis (PCA), which refers to conformational dynamics or essential motions related to the free energy landscape observed in the MD simulations. ProDy was also used for Gaussian network modeling to predict the hinge residues that play a key role in the essential motions of proteins.

Secondary structure analysis and related visualizations were performed by using the ggstride package (https://github.com/ugerlevik/ggstride/), which was developed for the first time to be used in this study. ggstride uses STRIDE (Frishman & Argos, 1995) for secondary structure assignment, bio3d (Grant et al., 2006) for structure and trajectory evaluations, and ggplot2 (Wickham, 2016) and ggpubr (Kassambara, 2020) for visualizations. This whole procedure was performed in R (v4.0.3) (R Development Core Team, 2020).

In PCA and secondary structure analysis, repeat 1 and 2 trajectories were merged while obtaining the observed conformations and secondary structure percentages. Only the last 75 ns of each trajectory was used in RMSF, PCA, secondary structure, and hinge analyzes as these parts had the most stable RMSD levels (Fig. 1) indicating that the structures had reached the energetic minimum.

Figure 1 Root-mean-square deviation (RMSD) analysis.

Backbone RMSD comparison between molecular dynamics simulations of wild-type and changed M1AP structures. R1 and R2 indicates “Repeat 1” and “Repeat 2” trajectories, respectively.

Analysis results were visualized using the ggplot2, ggpubr and gg.gap (https://github.com/ChrisLou-bioinfo/gg.gap) packages in R, and the matplotlib package (Hunter, 2007) in python (v3.7.6) (van Rossum & Drake, 2009). Structural visualizations were performed by using VMD.

Results

Evolutionary conservation

We observed that all variant sites except Ser50 were conserved among all orthologs examined (Fig. S4). Ser50 was also highly preserved by changing to a Cys only in the cat (Felis catus, UniProt ID: A0A337RXR8). This suggests that all these variants may have significant effects on the structure and function of meiosis 1-associated protein (M1AP). Therefore, all of them were further investigated using molecular dynamics (MD) simulations.

Molecular modeling and validation

Twenty-six models for the M1AP structure were built via the following algorithms with automatically selected templates: GalaxyWeb (five models), I-TASSER (five models), Phyre2 (one model), PRIMO (four models), RaptorX (one model), Robetta (five models) and Robetta-domain (five models). Although the properties regarding to the templates such as identity and coverage (Tables S1 and S2) indicate that the M1AP structure is difficult to model, it was possible to model the entire structure as the templates covered different parts of the M1AP and as utilizing the ab initio modeling. However, we required a comprehensive validation procedure as follows to proceed with the simulations reliably.

Firstly, we selected the best model from each algorithm by comparing their quality assessment scores (Table S3) to simplify benchmarking the models of the different algorithms. Remarkably, the models generated by the same algorithm were mostly similar to each other, except for the Robetta and Robetta domain models, which were more diverse than the models of the other tools (Fig. S2). As a result, model 1 from GalaxyWeb, I-TASSER, PRIMO and Robetta-domain, and model 5 from Robetta came forward for further benchmarking. Furthermore, among the single models of RaptorX and Phyre2 and the selected models of other tools, the top three models were from GalaxyWeb, Robetta and Robetta-domain (see “Initial Models” in Table 2). Since the GalaxyWeb model had the highest Ramachandran plot (PROCHECK) scores and the Robetta-domain model had slightly better scores than the Robetta model, we retained the GalaxyWeb and Robetta-domain models to continue with them as the two best models.

Instead of basic and static algorithms, we compared the two best models based on the observation of their folding stability over 100 ns of molecular dynamics (MD) simulations. We used the backbone root-mean-square deviation (RMSD) as an indicator of the stability of predicted protein folding. As shown in Fig. 2A, the GalaxyWeb model was quite stable after a short time whereas the Robetta-domain model continued to deviate throughout the entire simulation period. In addition, the quality scores of the models at the end of the 100 ns simulations differed only slightly from the initial structures, as expected (see “Further Models” in Table 2). Based on the RMSD results, we finally chose the GalaxyWeb model to examine the effects of variants. This M1AP model is available in the PMDB Protein Model DataBase (http://srv00.recas.ba.infn.it/PMDB/) (Castrignanò et al., 2006) under the accession identifier PM0083500.

Figure 2 Model details.

(A) Backbone root-mean-square deviation (RMSD) comparison between molecular dynamics simulations of GalaxyWeb and Robetta-domain models. (B) Overall view of the M1AP model. (C) The 2D topology of the M1AP model, including the localizations of variants. (D) The largest cleft (orange) in the M1AP structure and localizations of variants on the 3D model.

Since the AlphaFold2 (AF2) revolutionized the field of de novo protein modeling during the revision process of this article (Jumper et al., 2021; Thornton, Laskowski & Borkakoti, 2021; Varadi et al., 2021), we compared our final model with the AF2 model as an additional validation. As shown in Fig. S1, our model and AF2 model were quite similar (RMSD = 8.42 Å). In addition, the AF2 model has similar level of quality (i.e., better VERIFY3D and PROVE but lower ERRAT, PROCHECK and WHATCHECK scores) with our final model (see “Comparison Model” in Table 2). Because of this independently obtained resemblance of our GalaxyWeb model to this revolutionary de novo structure, we might infer that it is one of the best structures that can be computationally obtained by present knowledge.

Variant models were evaluated to see if they retained quality. As shown in “Variant Models” in Table 2, their qualities were quite similar to those of the wild-type near the end of the simulations (i.e., 450th ns), pointing out the validity of the simulation results.

Structural details

The final M1AP model has a globular shape (Fig. 2B) and involves 13 α-helices and 11 β-sheets. The detailed topology including the secondary structures generated by PDBsum is shown in Fig. 2C. Moreover, the model has a large cleft (volume: 11,448 Å3) shown in Fig. 2D. Upon observations of the absence of M1AP in the nucleus and impaired meiosis in the M1AP deficiency, Arango et al. (2013) theorized that M1AP might indirectly regulate meiosis progression by binding and activating a cytoplasmic protein that would later translocate into the nucleus. If this suggestion is correct, this large cleft in M1AP would be the most probable binding site for this theorized protein. Other clefts are not shown because they were significantly smaller (i.e., 3272 Å3, 2207 Å3, and 1507 Å3) and less likely to be suitable for ligand binding (Laskowski et al., 1996).

The locations of missense variants are shown on the tertiary structure of M1AP in Fig. 2D (also visible in the secondary structure topology of M1AP in Fig. 2C). Moreover, Ser50 and Leu430 are in α-helices; Gly317 is in a loop close to an α-helix; Pro389 is at the edge of a β-sheet; and Arg266 is in a surface-exposed loop. On the other hand, Pro389 and Leu430 are part of the largest cleft while Gly317 is close to it (Fig. 2D).

Impacts of the variants

Folding, compactness and solvent accessibility

The most stable part of the RMSD was the last 75 ns of each 500 ns trajectory, indicating that the simulations reached equilibrium (Fig. 1). Taking this into account, backbone deviation was lower in variants p.Ser50Pro (both repeats) and p.Arg266Gln (only repeat 1) and slightly higher in p.Gly317Arg (only repeat 1) than in wild-type whereas there were no remarkable changes in other variants (Fig. 1). Of note, a rapid and sharp increase was observed in the p.Leu430Pro variant before gradually decreasing and stabilizing. Combined with the radius of gyration (Rg) pattern (Fig. 3), it can be inferred that the structure partially unfolded and refolded during this time.

Figure 3 Radius of gyration (Rg) analysis.

Rg comparison between molecular dynamics simulations of wild-type and changed M1AP structures. R1 and R2 indicates “Repeat 1” and “Repeat 2” trajectories, respectively.

Rg was slightly lower in variants p.Arg266Gln (both repeats), p.Gly317Arg (only repeat 2) and p.Leu430Pro (both repeats), indicating that the protein was more compact upon these amino acid changes (Fig. 3). Rg of other variants did not differ from the wild-type. In any variant, we did not observe any critical solvent accessible solvent area (SASA) differences with wild-type (Fig. 4), which might be expected upon compactness changes.

Figure 4 Solvent accessible surface area (SASA) analysis.

SASA comparison between molecular dynamics simulations of wild-type and changed M1AP structures. R1 and R2 indicates “Repeat 1” and “Repeat 2” trajectories, respectively.

Stability and H-bonds

Although loss or gain of structural stability could be expected in line with RMSD and Rg changes, there were no significant differences in stability of any variant protein (Fig. S5). The number of H-bonds was also not different in any variant (Fig. S6), supporting the stability of the proteins to remain unchanged.

Local interactions

To understand how the local environment is affected by the amino acid changes, we analyzed the alterations in the H-bonds and salt bridges around 10 Å of each variant site. There were minor differences in the number of local H-bonds from the wild-type only in p.Ser50Pro (as decrease in both repeats), p.Arg266Gln (as increase in only repeat 2) and p.Leu430Pro (as decrease in only repeat 2) variants (Fig. S7). On the other hand, there were significant alterations in local salt bridges as mostly loss in the p.Arg266Gln (both repeats) and p.Pro389Leu (mostly repeat 1) whilst mostly gain in the p.Gly317Arg (mostly repeat 1) and Leu430Pro (both repeats) variants (Fig. S8). In addition, there were both type of salt bridge changes in the p.Ser50Pro variant.

Flexibility, hinge and essential dynamics

The conformational spaces observed in all variants were significantly different from the wild-type (Fig. 5), indicating altered essential protein dynamics and energetic minima of backbone conformations. In line with this observation, there were critical root-mean-square fluctuation (RMSF, indicating residue flexibility) changes in at least one repeat of each variant as pointed out by the yellow stars in Fig. 6, and hinge site (zero flexibility regions) differences as indicated by yellow and red colors in Table S4.

Figure 5 Principal component analysis (PCA).

PCA comparison between molecular dynamics simulations of wild-type and changed M1AP structures. The first two principal components are presented.

Figure 6 Root-mean-square fluctuations (RMSF) analysis.

Cα RMSF comparison between molecular dynamics simulations of wild-type and changed M1AP structures. Yellow stars indicate regions that differ from the wild-type. R1 and R2 indicates “Repeat 1” and “Repeat 2” trajectories, respectively.

Secondary structures

Ser50 and Leu430 were found in α-helices (Figs. 2C and 2D) and changed to Pro, which is one of the amino acids known as secondary structure breakers. Consistent with this expectation, both variants disrupted their α-helices, making them turn and 3(10)-helix structures (Fig. 7), which are less stable than α-helices. In p.Arg266Gln, the variant region was more stable as α-helix and turn rather than a coil observed in wild-type (Fig. 7). Furthermore, these were the most outstanding secondary structure changes at variant sites. However, there were less significant changes in other parts of the protein as well (Fig. S9).

Figure 7 Secondary structure analysis.

Secondary structure comparison of the variant sites between molecular dynamics simulations of wild-type and changed M1AP structures. Percentage refers to the percentage of the last 75 ns of simulation during which the amino acids were adopting each one of the secondary structure conformations.

Discussion

In this study, five non-obstructive azoospermia (NOA)-associated missense variants were individually analyzed for their impact on the meiosis 1-associated protein (M1AP) structure by performing molecular dynamics (MD) simulations. Moreover, none of the variants led to general misfolding or loss of stability of the M1AP structure. However, these amino acid changes severely disrupt the conformational dynamics of the protein as a result of altered flexibility, hinge and secondary structure properties due to the local interactions, particularly salt bridges near variant sites. These results were in line with our expectations based on the literature knowledge we had. Specifically, findings by Wyrwoll et al. (2020) and Arango et al. (2013) helped to shape these expectations and to propose the following hypothesis. Firstly, Arango et al. (2013) hypothesized an interaction partner for M1AP that goes later into the nucleus to regulate meiosis since they observed that M1AP deficiency cause disrupted meiosis but there is no M1AP presence in the nucleus. Considering our MD results on the conformational dynamics, these variants have a huge potential to disrupt such protein–protein interactions, which might relate them with the underlying mechanism of NOA. On the other hand, Wyrwoll et al. (2020) observed that these variants were bi-allelic or homozygous in the infertile individuals suffered from NOA whereas a heterozygous carrier was fertile and healthy. Thus, such a pathogenicity mechanism seems only possible when the wild-type M1AP is deficient in the cell. This indicates that one copy of the variant M1AP might not be adequate to cause NOA by sufficiently disrupting the related cellular function. Consequently, instead of being variant of uncertain significance as previously concluded in Wyrwoll et al. (2020), these variants might cause NOA when homozygous or bi-allelic. Of note, we propose this hypothesis based on our results and poor literature knowledge. Therefore, all these inferences should be tested with experimental structural and functional studies.

This study has limitations as described in the following. Even though M1AP is known to play an essential role in spermatogenesis (Arango et al., 2006, 2013), there is no further knowledge in the literature about its structure, functions, interactions, regulation and modifications. Therefore, we are currently unable to interpret the impacts of the alterations observed in MD simulations on upstream/downstream cellular functions, particularly through interactions of M1AP with others that could associate our results with the pathogenic mechanisms of NOA. However, there are many other studies that show similar structural distortions observed in different proteins (e.g., in secondary structures, flexibility of regions, hinge residues and essential dynamics) are frequently associated with diseases in a causative manner (Zheng, Hitchcock-DeGregori & Barua, 2016; Amir et al., 2019; Elghobashi-Meinhardt, 2020; Martínez-Archundia et al., 2020; Mohammad et al., 2020; Shuaib et al., 2020). Therefore, these M1AP variants may give rise to NOA considering their effects on protein structure.

Conclusions

Our aim in this study was to understand how non-obstructive azoospermia (NOA)-related missense variants previously reported in meiosis 1-associated protein (M1AP) gene affect the protein structure of M1AP. For this purpose, we firstly modeled the protein structure of M1AP, which has not been studied sufficiently in the literature despite its known importance in meiosis. We tried to obtain the best model we could by applying comprehensive validation procedures to existing methodologies. Next, we utilized molecular dynamics simulations, a robust method to find out how M1AP structure and dynamics are affected by the variants. Our results show that these variants significantly alter the conformational dynamics of M1AP by influencing local interactions and so flexibility, hinge and secondary structure patterns. Therefore, these variants might be causative for NOA when present homozygously or bi-allelically in individuals instead of being heterozygous, as supported by observations in Wyrwoll et al. (2020). However, more studies focusing on M1AP structure, function, regulation and interactions are needed to understand how these variants may affect cellular processes to cause NOA. Although we provide crucial insights about the pathogenicity mechanisms in the M1AP-associated NOA causing male infertility, our results and inferences must be tested by in vitro and in vivo studies.

Supplemental Information

Supplemental Information 1 AlphaFold2 (AF2) model comparison.

(A) M1AP model of AF2. Blue color refers to ≥70% confidence while others indicate lower confidence. (B) Structural alignment of GalaxyWeb (grey) and AF2 (orange) M1AP models, including root-mean-square deviation (RMSD) value.

Click here for additional data file.

Supplemental Information 2 Structural alignments of each model of multiple model generating algorithms to its model 1.

Root-mean-square deviation (RMSD) values are included. Grey color indicate model 1.

Click here for additional data file.

Supplemental Information 3 Potential energy analysis.

The energy pattern of each system during the entire simulation procedure.

Click here for additional data file.

Supplemental Information 4 Evolutionary conservation analysis.

Alignment of the human M1AP protein to its five orthologs. Yellow stars with labels indicate the positions of the variants examined in this study.

Click here for additional data file.

Supplemental Information 5 FoldX folding stability analysis.

Stability comparison between molecular dynamics simulations of wild-type and changed M1AP structures. R1 and R2 indicates “Repeat 1” and “Repeat 2” trajectories, respectively.

Click here for additional data file.

Supplemental Information 6 Overall H-bond analysis.

Comparison of the number of all H-bonds between molecular dynamics simulations of wild-type and changed M1AP structures. R1 and R2 indicates “Repeat 1” and “Repeat 2” trajectories, respectively.

Click here for additional data file.

Supplemental Information 7 Local (within 10 Å of the variant site) H-bond analysis.

Comparison of the number of local H-bonds between molecular dynamics simulations of wild-type and changed M1AP structures. R1 and R2 indicates “Repeat 1” and “Repeat 2” trajectories, respectively.

Click here for additional data file.

Supplemental Information 8 Local (within 10 Å of the variant site) salt bridge analysis.

Comparison of the local salt bridges between molecular dynamics simulations of wild-type and changed M1AP structures.

Click here for additional data file.

Supplemental Information 9 Secondary structure analysis.

Secondary structure comparison between molecular dynamics simulations of wild-type and changed M1AP structures. Percentage refers to the percentage of the last 75 ns of simulation during which the amino acids were adopting each one of the secondary structure conformations.

Click here for additional data file.

Supplemental Information 10 List of templates assigned for M1AP by the structural modeling algorithms.

The methods used to obtain the structures are given with resolution (the smaller the better) and Ramachandran outliers (the smaller the better) indicating the quality of the structures.

Click here for additional data file.

Supplemental Information 11 BLAST alignment details of the templates automatically selected by the structural modeling algorithms.

Click here for additional data file.

Supplemental Information 12 Quality assessments of the models built by multiple model generating algorithms from five different algorithms.

(A) GalaxyWeb, (B) I-TASSER, (C) PRIMO, (D) Robetta and (E) Robetta-domain.

Click here for additional data file.

Supplemental Information 13 Hinge site analysis.

Hinge site comparison between molecular dynamics simulations of wild-type and changed M1AP structures. Same colors refer to matching regions. Red and yellow colors indicate the different regions from that found in the wild-type structure.

Click here for additional data file.

Molecular dynamics calculations reported in this study were performed partly in TUBITAK ULAKBIM, High Performance and Grid Computing Center (TRUBA), and partly in Department of Biostatistics and Bioinformatics, Institute of Health Sciences, Acıbadem Mehmet Ali Aydınlar University.

Additional Information and Declarations

Competing Interests

Author Contributions

Data Availability

The authors declare that they have no competing interests.

Umut Gerlevik conceived and designed the experiments, performed the experiments, analyzed the data, prepared figures and/or tables, authored or reviewed drafts of the paper, and approved the final draft.

Mahmut Cerkez Ergoren conceived and designed the experiments, authored or reviewed drafts of the paper, co-author of Wyrwoll et al. (2020), started this study, and approved the final draft.

Osman Uğur Sezerman conceived and designed the experiments, analyzed the data, authored or reviewed drafts of the paper, and approved the final draft.

Sehime Gulsun Temel conceived and designed the experiments, authored or reviewed drafts of the paper, co-author of Wyrwoll et al. (2020), started this study, and approved the final draft.

The following information was supplied regarding data availability:

The generated and validated structural model of M1AP is available in the Protein Model Data Base (http://srv00.recas.ba.infn.it/PMDB/): PM0083500.

The data generated by molecular dynamics simulations and the related configuration files are available at Zenodo: Gerlevik, Umut, Ergoren, Mahmut Cerkez, Sezerman, Osman Ugur, & Temel, Sehime Gulsun. (2021). All-atom Molecular Dynamics Simulations of Meiosis 1-associated protein (M1AP) to Investagate the Impact of Known Missense Mutations Associated with Male Infertility through Non-obstructive Azoospermia [Data set]. Zenodo. https://doi.org/10.5281/zenodo.5811977.

All codes used in the analysis are available at GitHub: https://github.com/ugerlevik/M1AP_analysis.

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
