# Peer review of "Structural analysis of M1AP variants associated with severely impaired spermatogenesis causing male infertility"

_PeerJ, doi:10.7717/peerj.12947_

## Round 0.1 · original submission · Major Revisions

Please address concerns of the reviewers and amend your manuscript accordingly.

·

Basic reporting

Good English, Sufficient literature provided, Professional article and row data sharing.
The article is well witten and coclusions supported by data analysis

Experimental design

Thde paper is well conceived, interesting and conclusion are supported by investigations

Validity of the findings

Results will add the the knowledge in the field. One major comment is whether the authors can reinforce final conclusions possibly adding referring to experimental findings even by other researchers, if any.

Data are mainly computational and conclusions are mainly derived from theoretical observations.

Additional comments

One general comment is whether the authors can reinforce final conclusions possibly referring to experimental findings even by other researchers, if any.

·

Basic reporting

1) The manuscript has a significant amount of typos and redaction flaws and the English language needs to be improved to ensure that the readers can clearly understand your work. For example, in lines 93-94:
“MethodsA comprehensive whole exome sequencing and variant prioritization study (Wyrwoll et al., 2020) have firstly shown that the biallelic missense...”
A line break is missing after “Methods” and “has previously” would be more appropriate than “have firstly”.
The manuscript would greatly benefit from being reviewed by someone who is proficient in English and familiarized with your research subject, like a colleague or a professional editing service.

2) In Methods the authors list in Table 1 the five missense variants individually. According to Wyrwoll et al., 2020, the different individuals with infertility presented the following bi-allelic variants:
p.Ser50Pro and p.Leu430Pro, in compound-heterozygosity
p.Gly317Arg in compound-heterozygosity with a loss-of-function variant p.Trp226LeufsTer4 reported as pathogenic when homozygous
p.Arg266Gln in compound-heterozygosity with a loss-of-function variant p.Trp226LeufsTer4 reported as pathogenic when homozygous
p.Pro389Leu as homozygous in an infertile individual, and as heterozygous in his fertile brother.
In order to avoid confusions about the bi-allelic combinations of these varians, I suggest the authors consider presenting this information about the zygosity of the five VUS, either in Table 1 or in the Introduction section. I also suggest to add the scores of the In Silico Predictions (PolyPhen-2/SIFT/MutationTaster) assessed by Wyrwoll et al., 2020

3) Since the pathogenicity of these five variants has still not been confirmed by functional experiments, and four of them are biallelic with other Pathogenic variants, I would like to suggest to avoid using the term “mutation”. Use instead “variant” or “change”, as recommended by the Human Genome Variation Society (HGVS) http://varnomen.hgvs.org/bg-material/basics/. However, this is subject to the preferences of the Journal.

4) I appreciate the effort of the authors in preparing the figures and providing all the supplemental information, however I find some of the figures and tables are hard to follow, inappropriate resolution and their legends do not provide enough explanation. For the supplementary figures, I could not find any legends.

To mention some things that could be improved in figures:

-Figure 2:
In panels (B) and (D) the text is hard to read. Is always better to avoid superimposing text over the structures as much as possible, prefer to place text outside and point to the amino acids with arrows. The sticks of the lateral chains are barely distinguishable from the cartoon representation. It is neither clear where the variants are in the predicted secondary structures in (B) nor if they are part of or close to the putative cleft in (C).

-Figures 3-6 should ideally have consistent start-end values in the Y axes of their panels.


-Figure 8:
The text is too small, the residue indexes can not be read. It is not clear to me what is the meaning of “Percentage of secondary structure of the residues near each mutation site..”. Does it refer to the percentage of the total time of simulation during which the amino acids were adopting each one of the secondary structure conformations?


-Figure_S1:
It is not clear to me which models are being aligned in each panel, do the colours have a meaning?. Also, changing the point of view of the structures between the panels is visually quite disorienting. It is better if the same view is consistently used in all the panels, unless the authors were trying to show different aspects, in which case it should be explained in a figure legend.

-Figure_S2:
For a more clear identification of where the variants are in the multiple sequence alignment in Figure_S2, the corresponding positions should be highlighted. This could be done with a symbol on top of it or a box enclosing the column, for example.

-Supplementary material 2
It is not clear what is the meaning of the different colours

5) For a better narrative in concordance with the order of the sections in Methods, I recommend starting the Results with the “Evolutionary conservation” section in first place, and then move on to the structural analysis.


6) The abbreviations used in the manuscript should be introduced on first usage in the Abstract, and again if/when they appear in the main body of the document. Please, explain NOA also in the introduction.

7) The reference for UniProtKB (lines 108-109) should be introduced instead in the “Evolutionary conservations” section, where it is first mentioned.

Experimental design

Five variants were previously observed in bi-allelic combinations in male individuals diagnosed with infertility and reported as variants of uncertain significance (VUS) for this phenotype (Wyrwoll et al., 2020).
In the present work, the possible impact of these five missense variants, observed in the Meiosis 1-Associated Protein (M1AP), is being investigated using protein structure homology modelling followed by molecular dynamics simulations.

I would like to mention three key points that need to be addressed in the experimental design of this work.


1) According to what Wyrwoll et al., 2020 mention in their report:
To gain insight into the function of M1AP and better assess the relevance of identified variants, we pursued two strategies. First, we attempted to model M1AP’s 3D structure. However, because of the lack of information on M1AP and comparable 3D structures, it was not possible to achieve a reliable prediction (BLAST results for sequence of UniProt: Q8TC57 are below 30% sequence identity to known protein structures, details in Supplemental Methods).

In the present work, the Table 2 in “Results” lists the experimentally determined structures selected as temples by the modelling programs, as described in the “Molecular modelling and validation” section. However, the sequence identity between the target sequence and this structures, along with the experimental methodology (X-ray, NMR, cryo-EM) and quality (i.e. RMSD, R-factors when applicable) of these structures are not being provided. This information is crucial when performing homology modelling, as it determines, along with the model quality assessment, to which extent the final models can be trusted with enough level of detail for further computational analysis. Particularly the calculations of stability, amino acid contacts, surface cavities and molecular dynamics, like the ones performed afterwards by the authors, require as input high quality structures (in general atomic resolution with values <3Å) as their estimations are very sensible to lateral chains orientations and interactions.
The authors should properly describe all this information of the experimental structures chosen as start templates along with the regions of the target sequence effectively covered by the templates and their oligomeric state, paying particular attention to coverage and similarity/identity of the positions affected by the missense variants and its surroundings.

I recommend the authors to go through this recent very comprehensive and clear review for good practises in protein modelling:
Haddad Y, Adam V, Heger Z (2020) Ten quick tips for homology modeling of high-resolution protein 3D structures. PLoS Comput Biol 16(4): e1007449. https://doi.org/10.1371/journal.pcbi.1007449


2) In the “Molecular modelling and validation” section, it is not clear how the model for the structures with the variants were generated and validated. Accompanied by this, in the Results sections there is no mention of the quality assessment of these variant models.


3) In the “System preparation section”, some of the key parameters for a molecular dynamics simulation are not mentioned:
-Neither the barometer nor the Gamma parameter from the Langevin thermometer are provided.
-The criteria adopted to decide when the system was equilibrated and ready for the production MD is not clear. Did the energy of the system improve after the relaxation?
-The xyz dimensions (line 128) of the system box lack a unit of measure.

Validity of the findings

I acknowledge the huge work that the authors have done for obtaining the model structures and in performing subsequent molecular dynamics simulations. However, there are multiple weak points where the authors fail in supporting the validity of their findings:

1) In Results, “Molecular modeling and validation” section, lines 206-208, the authors say:
“Although PROCHECK and WHATCHECK scores of this final model were slightly decreased, the VERIFY3D, ERRAT and PROVE scores were significantly improved after the 100 ns MD simulations as shown in Table 3”.

It is true that VERIFY3D, ERRAT and PROVE scores have improved for the model after the MD simulations, but still:
a) ERRAT AOQF of 86.99% is lower than 91% (bad quality),
b) PROVE BOPA 6.5% is higher than 5% (bad quality),
c) PROCHEK and WHATCHECK give bad scores to the GalaxyWeb model and to the model after the MD simulations

The authors are giving a final assessment of “Good” to the “improved” model, based only in one of the five scores (VERIFY3D), disregarding the bad scores of the remaining four.


This lack of rigurosity, along with the absence of proper information about the PDB structures used as templates, provides a dubious support to the reliability of the models for all the further structural analysis carried out by the authors
Also, as I mentioned before, the quality assessments for the structural models with variants, are not presented.


2) In Results, “Molecular modeling and validation” section, lines 184-186, they authors say:
The multiple models generated by the same tool (i.e., GalaxyWeb, I-TASSER, PRIMO,
Robetta and Robetta-domain) were highly similar to each other (see the structural
alignments in Fig. S1).
As I mentioned before in the Basic Reporting section, Fig. S1 lacks a figure legend and is poorly informative regarding which models are being superposed in each panel. Overall, except maybe in panel (A), the structural superpositions in this figure clearly show differences between the structures, there is an abundance of loopie regions adopting multiple conformations and hence conferring many possibilities of orientations to the alpha helices and beta strands that they connect. Nevertheless, it is not accurate to rely simply on the visual observation of the superimposed structures, the degree of structural similarity has to be properly quantified using, for example, RMSD between the aligned structures.


3) In Results, “Structural details” section, lines 213-215, the authors suggest:
“Although there is currently no known interaction in the literature, this cleft of this underrecognized protein is suitable for being a binding site for a ligand such as DNA”
The authors are being too speculative without further discussing in what evidence they base their suggestion. What made them think that the putative cleft is “suitable for being a binding site for a ligand such as DNA”?

4) The RMSD (Figure 3) , RMSF (Figure 6) and Foldx ΔG of unfolding (Figure S3) show high values and also high oscillations between the different time steps of the simulation, and between the step 1 and step 2. This tells that the structures are experiencing big movements and have high and fluctuating energies, even when only looking at the wild-type models. This could be an indicative that the structures were not correctly stabilized in the pre-production MD simulation, but also that the structural quality of the models is not good.

5) Based on their Results, the authors discuss (lines 291-293):
“Instead of being variant of uncertain significance as previously concluded in Wyrwoll et al. (2020), our findings validated that these variants are the lead candidates for NOA”
The authors should be very cautious with the extent of support their results can give to their discussions and conclusions (lines 291-293 and 308-310). I strongly suggest reviewing this and reformulating. It is recommended that computational analysis can be used for adding support and prioritizing candidate variants and exploring possible molecular mechanisms, but in order to validate and confirm functional effect and pathogenicity, this has to be combined with functional experimental assays.

In case they have not done so, I recommend the authors to go through this two resources:
-The ACMG guidelines: Standards and guidelines for the interpretation of sequence variants: a joint consensus recommendation of the American College of Medical Genetics and Genomics and the Association for Molecular Pathology, Genet Med 17, 405–423 (2015), https://doi.org/10.1038/gim.2015.30
-Guidelines for investigating causality of sequence variants in human disease, MacArthur et al., 2014, Nature 508, 469–476 (2014), https://doi.org/10.1038/nature13127

Additional comments

I acknowledge the enormous effort the authors have invested, considering that M1AP represents a challenge, since it lacks elucidated protein structures. This is a very interesting and important protein to study, the elucidation of how the molecular mechanisms of its normal function are being altered by variants leading to male infertility, is urgently needed, as this knowledge could hopefully add valuable information for developing treatments in the future.

However, the present work lacks rigurosity and fails in key points to ensure the validity of the author’s findings: the methods are not clearly explained to ensure reproducibility and the structural models here obtained fail the quality assessments for being used in the subsequent structural analysis. Unfortunately, not accomplishing these fundamental points makes this manuscript, in its current state, not appropriate for its publication.

---

## Round 0.2 · accepted · Accept

All critiques were carefully addressed and the manuscript was amended accordingly. Therefore, the revised version is acceptable now.